# The meritocracy trap: Early childhood education policies promote individual achievement far more than social cohesion

Katarzyna Bobrowicz[1,2]*, Pablo Gracia[3,4], Ziwen Teuber[1], Samuel Greiff[5]

1 Department of Behavioural and Cognitive Sciences, University of Luxembourg, Esch-sur-Alzette, Luxembourg, 2 Psychology and Neuroscience of Cognition, Faculté de Psychologie, Logopédie et Sciences de l'Éducation, Université de Liège, Liège, Belgium, 3 Department of Sociology, Universitat Autònoma de Barcelona, Bellaterra, Spain, 4 Centre d'Estudis Demogràfics, CED-CERCA, Bellaterra, Spain, 5 School of Social Sciences and Technology & Centre of International Student Assessment, Technical University Munich, Munich, Germany

* katarzyna.a.bobrowicz@gmail.com

## Abstract

Governments worldwide have reformed early childhood education (ECE) to equip young people with competitive skills for an increasingly specialized workforce. These reforms have coincided with a widespread acceptance of meritocratic beliefs holding that talent and effort, rather than uncontrollable factors (e.g., luck, social context), determine individuals' lifetime success and achievement. This study examines whether recent ECE reforms may have promoted an economic meritocratic mindset that favors skills linked to individual competition for future achievement. Data came from a total of 92 documents published between 1999 and 2023, including ECE advisory reports from international organizations and government-endorsed ECE curricula from 53 countries across Africa, the Americas, Asia, Europe, and Oceania. A step-by-step thematic analysis was conducted through combining qualitative text coding with statistical analyses applied to the emerging themes. Findings show that: (1) while experts and policymakers recognized the importance of ECE access and quality, they defined social cohesion primarily through economic indicators; (2) ECE documents prioritized cognitive skills and –mostly among international organizations– socioemotional skills as key for individual achievement, but citizenship skills were largely omitted; (3) individual agency and responsibility within ECE contexts were defined as central to educational and lifetime success, while uncontrollable factors (e.g., intergenerational transmission of advantage, family origin) were largely neglected; (4) both international organizations and governments strongly embraced an economic meritocratic mindset in ECE, implying that life outcomes mainly depend on talent and effort, obscuring the role of support and solidarity from peers, relatives, communities or institutions. Overall, this study suggests that ECE reforms have globally reinforced the pitfalls of meritocracy by promoting educational policies that

**Data availability statement:** All relevant data are within the manuscript and its Supporting Information files.

**Funding:** The author(s) received no specific funding for this work.

**Competing interests:** The authors have declared that no competing interests exist.

prioritize competition over cooperation, individualism over solidarity, and the widespread notion that talent and effort, rather than uncontrollable factors such as luck or social context, determine individuals' lifetime success in society.

## Introduction

In recent decades, government investments in Early Childhood Education (ECE) have significantly increased to prepare future workers for rapidly unfolding social and technological transformations, and to equalize children's life chances [1–4]. Recent educational reforms have resulted in a global increase of ECE access, paralleled by a surge in wealth and skill-based economic inequalities [5]. Despite the well-documented persistent inequalities in children's life opportunities across social groups, research shows that *meritocratic beliefs* have grown significantly in recent decades [6]. Meritocracy is a moral and economic mindset guided by two key principles: (1) that citizen's worth is primarily a reflection of individual economic achievement and labor market position; and (2) that individual achievement is a byproduct of 'talent' and 'effort', materialized into human capital [6–8]. Meritocratic values have been found to have strong public relevance across North America and Western Europe, and they seem to have become more relevant in other regions, too, such as East Asia and Latin America [7–13]. Given that hegemonic principles shape educational agendas and policies (e.g., the goal of prioritizing individual gains and competition rather than equality and cooperation) [14], examining whether ECE reforms embody meritocratic principles allows to understand which skills and values international organizations and state governments are currently promoting.

The present study is, to our knowledge, the first exhaustive analysis of how recent ECE reforms link to the economic meritocratic mindset. Following previous literature [6–8], we argue that the economic meritocratic mindset (1) does not work for the achievement and well-being of all its members, only for those of certain societal groups, and (2) in turn, it promotes competition for limited resources at the expense of social cooperation, thereby threatening social cohesion [15–17]. Since education echoes dominant social and economic thought [18–19], and the last decades have indeed witnessed the rise of meritocratic views in society, policy decrees and early curricula around ECE may be widely embedded in the hegemonic economic meritocratic mindset discourses. Previous research suggests that school socialization can strengthen individual support for education-based meritocracy [20–21]. Other studies reveal that ECE practices are guided by country-wide policy documents [22] within a governance context that is not questioning rampant social inequalities to tackle social cohesion [23]. To date, however, research has lacked an approach to study the meritocratic mindset in ECE policies, with studies typically focusing on primary education [14,24], examining general curricular values in an explorative manner [25], and involving small and often regional samples (up to 13 countries; [25–28]). Additionally, the existing literature employed only one type of method (i.e., qualitative methods) [25,26], without actually focusing on the meritocratic mindset [29–31]. This calls for

new research combining various methodological approaches to examine from an international perspective the way official ECE documents may promote the economic meritocratic mindset.

Our study specifically contributes to the literature by examining whether key international and national level stakeholders in the formation of ECE curricula and educational plans may have favored certain skills linked to competition or individual achievement, as opposed to skills linked to empathy, citizenship, and collaboration. Critically, we analyzed whether ECE documents attribute enough importance to the role of uncontrollable external factors in individual outcomes. Examples of uncontrollable factors included a broad diversity of relatively random external events (e.g., experiencing an unexpected sickness or having good health; suffering from parental death or harsh parenting versus having healthy and highly supportive parents; experiencing a climate disaster during childhood), as well as socioeconomic and demographic factors (e.g., parental social class, neighborhood of residence, race/ethnicity) [7,8]. The data analyzed in the study came from a total of 92 documents published between 1999 and 2023. To capture macro-level factors promoted by key institutions at international and national levels with actual impact on actual ECE functioning, we focused on both ECE advisory reports from recognized international organizations (e.g., the World Bank, the OECD, the European Union), as well as ECE government-endorsed curricula from 53 countries across Africa, the Americas, Asia, Europe, and Oceania. We conducted a thematic analysis with a mixed-method approach that combined the identification of text content through qualitative identification of codes to create themes, followed by statistical analyses that examined the prevalence of the different emerging themes across the documents.

## Background

### Literature review

The concept of meritocracy was coined in the Western context during the 1950s, as an approach to social stratification wherein higher occupational status and income were conditional upon higher educational levels [32–34]. This approach would foster inequality by dividing society into "the blessed" and "the unblessed" [32] or "the winners" and "the losers" [8], while solidifying inequality of position through income, property, education and occupation under the disguise of "equality of opportunity" [32]. The second half of the 20th century witnessed a massive dissemination of meritocratic beliefs, coupled with two other critical concepts: neoliberalism and Human Capital Theory. Neoliberal state policies, including those concerned with ECE, are typically associated with moving the responsibility for key services such as educational and healthcare provisions away from the state and toward the private sector [35]. These policies stem from an assumption that education is a tool to improve state's economy [36], drawing on the Human Capital discourse. According to this perspective, human capital is strongly correlated with the economic growth of the nation [37] and individuals are viewed as active entrepreneurs rather than passive receivers of state-granted welfare [38]. Previous studies have argued that neoliberal policies promote Human Capital Theory to draw attention to solutions that emphasize individual responsibility for one's own education, training, and employability, away from tackling poverty and inequality through redistribution of wealth [39]. Neoliberalism does not promote social equality; quite the contrary – it is built on the notion that individuals are not equal, and some will win, while others will lose when competing for resources [40], much like in the economic meritocratic mindset [8]. Likewise, neoliberalism promotes values of self-reliance, autonomy, and independence as preconditions of self-esteem and self-worth, thus discarding the collective responsibility for the vulnerable in the society [38].

Previous literature illustrates how ideological processes underpin educational policies across the globe [14]. A longitudinal analysis of the "Starting Strong" series of international reports issued by the OECD showed that, over the last two decades, the rhetoric shifted from the appreciation of pedagogical diversity in the inaugural "Starting Strong I" decades ago to the current framing of such diversity as a threat to educational quality [41]. Across the globe, neoliberalism seems to have permeated the ECE context by promoting self-regulation-centered pedagogies [42] and child-centered discourse [43] that foster individualist narratives of success [44] and reinforce inequalities across children from different sociocultural backgrounds [45]. In the child-centered discourse, the child is seen as a self-governing, responsible individual capable of

initiative, persistence, and autonomous decision making (cf. the "Athenian child" in [46]), a notion that has been previously found in selected ECE curricula [31]. These individualistic values are closely aligned with the meritocratic mindset, wherein individuals, and children themselves, are responsible for their own successes and failures [42,46].

Furthermore, large-scale longitudinal evidence shows clear changes in educational policies in primary education in recent decades, such as a growing emphasis on students' individual development, employability, and adaptability to social change, but also a focus on democracy and education as a human right between 1965 and 1980 [14,24,47]. This literature has observed a shift from top-down teaching of prescribed, discipline-specific knowledge toward self-regulated learning and interdisciplinary knowledge construction [31,47]. This finding reveals a growing emphasis on students' adaptation to social change and labor market demands in educational contexts. Although these documents may indicate the rise of the economic meritocratic mindset through official ECE documents globally [14,47], previous studies have not tackled this matter exhaustively.

## The present study: theoretical framework and hypotheses

The present study targets the economic meritocratic mindset in ECE by analyzing a larger-than-ever country sample, using mixed methods applied to a thematic analysis. Based on previous literature [7,8], we define economic meritocracy as an ideology that assumes three principles, namely that (1) individual achievement relies merely on skill, understood as a product of talent and effort, and (2) individuals have a major agency over their own achievement, that is, (3) they do not depend on their group, community, or society for such achievement. In international expert and policymakers' fora, individual achievement, as defined in research on meritocracy, is typically a part of "lifetime success", an umbrella-term that involves individual economic capital, well-being, health, and social prospects [48,49].

Before moving to our hypotheses, it is necessary to situate this literature within the context of early childhood investment programs. In their seminal study, Carneiro and Heckman [50] argued that early childhood interventions would provide the highest returns for skill formation and that inequalities in outcomes started early in life, in fact, already within the family. Drawing on a life-cycle model that accounts for empirical evidence from programs like the *Head Start* in the Unites States [51], the relevant work has stressed the positive effects of early childhood investments in promoting high-school graduation, college attendance, earnings or avoidance of crime, particularly benefiting children from the most disadvantaged socioeconomic and demographic backgrounds. The economic idea of early childhood investments has been quickly popularized and spread over recent decades through –among others– intermediary organizations, institutional entrepreneurs, and think-tanks promoting the need of investing in early childhood to maximize and equalize returns, while reducing costs associated with later investments [52]. Certainly, programs such as the *Head Start* and similar interventions around the world partly succeeded in reinforcing values, such as citizenship, community, and equality. Yet, it is likely that policy makers and other top international stakeholders mainly employed ECE proposals that seek to foster –primarily– individual competition and success rather than other values, such as cooperation and empathy. It is, further, possible that different actors and stakeholders hold potentially contradicting views, for instance, regarding how ECE programs should be implemented in practice. To date, these questions remain unanswered. Our study contributes to these debates by analyzing whether ECE ideologies may have globally promoted competitive ideologies anchored into the economic meritocratic mindset, and whether these views have been consistent across different high-level actors involved in ECE systems and discourses.

Our study investigates whether the economic meritocratic mindset may be promoted in the ECE curricula worldwide by invoking the concepts of individual achievement and lifetime success. In an economic meritocracy, educational attainment is closely associated with economic gains and contributes to the accumulation of capital and power among well-educated individuals, while restricting life opportunities of the less educated [8]. The assumption that the meritocratic mindset may have underlain educational reforms is strengthened in standard life-cycle human capital models [e.g., 53,54]. In life-cycle human capital models, early education is seen as an investment in individual human capital and future economic returns,

wherein pupils compete to achieve skills that are differently rewarded in the market [1,18,19,55–57]. This competition between individuals is inscribed into the modern meritocratic mindset [58], captured by the metaphor of climbing the social ladder as a representation of upward social mobility [33]. Climbing up is a solitary, not a solidary activity. This strategy may overemphasize individual skill, effort, agency and control in one's own social mobility, while downplaying the contribution of the uncontrollable factors beyond individual control. Examples include the inheritance of genes and socioeconomic advantage, luck (i.e., good or bad fortune), contemporary economic demand on individual skills, economic valorization of one's own profession, and the support of other members of the community and the society [7,8,59,60]. These values, consequently, emphasize individual skills and agency, while obscuring the social forces that contribute to individual achievement. Since, in principle, educational policy documents guide country-wide pedagogical practices in ECE facilities [22], such documents may translate into instilling the meritocratic approach in young children.

We developed two levels of hypotheses, based on the following theoretical assumptions on how ECE stakeholders may promote economic meritocratic mindset. The first assumption of economic meritocracy (i.e., that individual achievement relies merely on skill and that skill is primarily a product of talent and effort) fails to account for aspects such as the socioeconomic context (e.g., current market and normative value of given skills in the labor market or society), as well as luck or misfortune. Accordingly, *educational stakeholders are expected to emphasize children's skills linked to individual lifetime economic success and associated with their market value, paying less attention to skills linked to solidarity and cooperation* (H1).

The second and third assumptions of economic meritocracy (i.e., that individuals have major agency over their own achievement, and that this achievement does not depend on educational support or social network) emphasize individual responsibility for one's own life outcomes, missing the role played by social and community contexts. Hence, *educational stakeholders are expected to emphasize children's agency and independence in lifetime success, without promoting individual awareness of the critical role that uncontrollable factors and support from others play behind individual achievement* (H2).

## Materials and methods

### Data sampling

This study adopted a modified six-phase latent thematic analysis [61,62], which typically consists of familiarization, coding, identifying themes, reviewing themes, defining and naming themes, and, finally, writing the report [61]. A typical six-phase thematic analysis is inductive, as the first phase – familiarization – involves an in-depth close reading of the text in search of potentially relevant data. However, our analysis combined deductive and inductive approaches. The examined themes were defined before analyzing the data, based on previous literature [7,8,58], and, thereafter, the indicators for each theme were identified inductively, based on a close reading of the text by the first author.

To define the sample ahead of analyses and guide our methodological choices, standards for Reporting Qualitative Research (SRQR) [63] and the Big Q Qualitative Reporting Guidelines (BQQRG; [64]) were strictly followed. First, documents on ECE authored by the OECD, the World Bank, and the EU were identified via the OECD Library, the World Bank's Documents & Reports site, and the EU Eurydice register, respectively (N = 11; none by the World Bank were deemed relevant). Fifty-three countries were selected for the search of the government-endorsed curricula, which resulted in 83 documents, two of which were dropped as they were authored by the UNESCO, not the representatives of a given country (N = 81). This resulted in a total of 92 documents.

The sample of the countries included in the analyses comprised both OECD Member Countries/Key Partner countries and non-Members (see Table 1). All documents in languages other than English were translated and cross-checked across Google Translate and DeepL. Full texts of works were retrieved with Google Search engine, LubSearch engine and the search engine of Bibiliothèque nationale du Luxembourg. An overview of the sources and child's ages represented therein is available in Table 1. Note that some countries were represented by more documents than one, and that

**Table 1. Overview of selected countries of study.**

| Continent | Country | Document | Child's age |
|---|---|---|---|
| **Africa (7% of African countries)** | Ethiopia | [65] | unspecified |
| | Kenya | [66] | unspecified |
| | South Africa** | [67] | 0 to 4 years |
| | Tanzania | [68] | 0 to 5 years |
| **Asia (20% of Asian countries)** | China (mainland)** | [69] | 3 to 6 years |
| | Hong Kong | [70] | kindergarten to secondary six |
| | India** | [71] | 0 to 3 years |
| | Japan* | [72] | 3 to 5 years |
| | Malaysia | [73] | unspecified |
| | | [74] | unspecified |
| | | [75] | unspecified |
| | Pakistan | [76] | 4–5 |
| | | [77] | unspecified |
| | the Philippines | [78] | 5 years onwards |
| | | [79] | 5 years |
| | Republic of Korea* | [80] | kindergarten/ young children |
| | Singapore | [81] | unspecified |
| | Taiwan | [82] | unspecified |
| **Oceania (14% of Oceanian countries)** | Australia* | [83] | kindergarten to 12 years |
| | New Zealand* | [84] | unspecified |
| **Europe (59% of European countries)** | Albania | [85] | 3 to 6 years |
| | | [86] | 0 to 6 years |
| | Austria* | [87] | up to 6 years |
| | Belgium* | [88] | 1 to 5 years |
| | | [89] | 3 to 6 years |
| | Bulgaria | [90] | 3 to 6 years |
| | | [91] | unspecified |
| | Croatia | [92] | unspecified |
| | | [93] | unspecified |
| | Czechia* | [94] | unspecified |
| | Denmark* | [95] | 0 to 6 years |
| | Estonia* | [96] | 6 to 7 years |
| | Finland* | [97] | unspecified |
| | France* | [98] | unspecified |
| | Germany* | [99] | 0 to 11 years |
| | Iceland* | [100] | unspecified |
| | Ireland* | [101] | unspecified |
| | Italy* | [102] | unspecified |
| | | [103] | unspecified |
| | | [104] | unspecified |
| | Latvia* | [105] | unspecified |

*(Continued)*

| Continent | Country | Document | Child's age |
|---|---|---|---|
| | Lithuania* | [106] | 6 years |
| | Luxembourg* | [107] | unspecified |
| | | [108] | unspecified |
| | Norway* | [109] | unspecified |
| | Poland* | [110] | unspecified |
| | Portugal* | [111] | unspecified |
| | Romania | [112] | 0 to 3 years |
| | Slovak Republic* | [113] | unspecified |
| | Slovenia* | [114] | 1 to 6 years |
| | Spain* | [115] | unspecified |
| | Sweden* | [116] | unspecified |
| | The Netherlands* (commercial only) | [117] | 0 to 6 years |
| | | [118] | 2 to 7 years |
| | United Kingdom (England and Northern Ireland)* | [119] | 0 to 5 years |
| | | [120] | unspecified |
| **Middle East (22% of Middle-Eastern countries)** | Jordan | [121] | unspecified |
| | Israel* | [122] | unspecified |
| | | [123] | unspecified |
| | Saudi Arabia | [124] | 0 to 6 years |
| | | [125] | 0 to 3 years |
| | Türkiye* | [126] | 0 to 3 years |
| | | [127] | 0 to 3 years |
| **North America (13% of North American countries)** | Canada (Prince Edward Island and Quebec)* | [128] | 0 to 4 years |
| | | [129] | 3 to 6 years |
| | | [130] | unspecified |
| | | [131] | 4 years |
| | Mexico* | [132] | unspecified |
| | | [133] | unspecified |
| | USA (commercial and governmental)* | [134] | 0 to 3 years |
| | | [135] | unspecified |
| | | [136] | unspecified |
| | | [137] | 3 to 5 years |
| | | [138] | 0 to 3 years |
| | | [139] | unspecified |
| | | [140] | 0 to 8 years |
| | | [141] | 0 to 5 years |
| | | [142] | 0 to 5 years |
| **South America (25% of South American countries)** | Argentina | [143] | unspecified |
| | Brazil** | [144] | unspecified |
| | Chile* | [145] | 0 to 5 years |

*(Continued)*

**Table 1.** (Continued)

| Continent | Country | Document | Child's age |
|---|---|---|---|
| **International** | EU | [146] | unspecified |
| | | [51] | unspecified |
| | | [53] | unspecified |
| | | [54] | unspecified |
| | | [147] | unspecified |
| | OECD | [148] | 0 to 5 years |
| | | [149] | unspecified |
| | | [1] | unspecified |
| | | [2] | 0 to 5 years |
| | | [48] | unspecified |
| | | [49] | unspecified |

Thirty-three out of OECD Member Countries and twenty other countries across six continents were considered. *OECD Member Countries. **Brazil, China, India, and South Africa are not OECD Member Countries but serve as OECD Key Partners (For details see https://www.oecd.org/about/members-and-partners/).

some were represented by governmental guidelines (e.g., Sweden) or commercial curricula (e.g., the Netherlands and the United States; Table 1). To situate the regional background of the researchers in the team, three co-authors of this manuscript were immersed in the Western European educational context throughout their education, and one co-author experienced both the Western European and the Eastern Asian educational contexts. We focused our analyses on the advisory reports of international organizations and the reformed early childhood curricula of 53 countries across six continents between 1999 and 2023 (see Fig 1 and Table 1). We ensured that both OECD Member Countries (33) and non-Members (20) were included. The selection of countries was guided by the availability of ECE policy documents.

## Empirical analyses

A mixed-method thematic analysis was applied to analyze the advisory reports and the policy decrees, as well as the ECE curricula. To investigate whether the international expert and policymakers' fora promoted the economic meritocratic approach, we analyzed relevant documents on ECE issued by the OECD and the European Union (for an overview see Table 1). Further, we examined policy decrees and early educational curricula. Across the investigated sources, we sought indications that children were taught to (1) recognize that not only skill, but also socioeconomic context and luck/misfortune underpin one's own and others' successes and failures, and (2) appreciate the reliance on others in their own success. In the analysis, we identified four themes, divided into twenty-eight indicators (see Table 2) to examine the outlook of the selected stakeholders on whether skills, individual agency and independence, uncontrollable events, and contribution of others matter for lifetime success.

Ahead of data analysis, a comprehensive literature search was carried out to examine which themes may map onto the research questions. Particular attention was paid to the works by Markovits [7] and Sandel [8], applying these works to the case of ECE reforms. Based on the core sources, the themes that capture what constitutes a meritocratic mindset were identified as (1) focus on the individual skill and control in lifetime success, (2) negligence of uncontrollable factors behind lifetime success, and (3) underappreciation of social interdependence for lifetime success (Phases 1 & 2: identifying the themes and defining and naming themes).

To investigate whether this mindset underpinned stakeholders' approach to ECE, and how it may be manifested in the selected sources, all sources were first read, and all possible indicators were noted down by Rater 1 (Phase 3:

 

**Table 2. Overview of key themes and indicators retained in the analysis.**

| Theme | Sub-theme | Indicator | Definition | Abbreviation |
|---|---|---|---|---|
| **Skills matter for lifetime success** | | Cognitive | Abilities and behaviors that involve higher-order cognitive processing, e.g., language, mathematics, categorization, attention, also in situations involving other agents, e.g., communication, collaboration. | COG |
| | | Socio-emotional | Abilities and behaviors that involve recognition and management of emotions, one's own and others, in both individual and group activities. | SOC-EMO |
| | | Physical | Abilities and behaviors that involve senses, movement and coordination and serve physical strength, communication (e.g., pointing and other gestures), and artistic development (e.g., dancing). | PHYS |
| | | Self-care | Abilities and behaviors that allow the individual to take care of one's own and others' health, well-being and safety. | SELF-CARE |
| | | Artistic | Abilities and behaviors that support the development of artistic awareness and expression. | ART |
| | | Citizenship | Abilities and behaviors that support individual participation in the community and the society. | CITIZEN |
| | | Achievement | Explicitly mentioned individual achievement, goals or accomplishment. | ACHIEV |
| **Individual agency and independence matter for lifetime success** | | Agency | Child's capacity to act on the physical and social environment. | AGENCY |
| | | Initiative | Child's capacity to seek and initiate contact, activities, and reactions of others. This includes such actions as asking, being spontaneous, creating, seeking information etc. | INITIATIVE |
| | | Confidence and self-efficacy | Child's feeling or belief in their own abilities and self-worth. Also child's feeling or belief in their own capacity to achieve goals. | CONFID |
| | | Persistence and perseverance | Child's propensity to keep to a certain plan or action until it is finished (or for an increasing amount of time on the way to the finish line). Also child's propensity to keep to a certain plan or action despite challenges, difficulties, and failures. | PERS |
| | | Decision making | Child's capacity to make choices and decisions with respect to plans, actions and goals. | DECIS |
| | | Effort | Implied or explicitly mentioned child's effort. | EFFORT |
| | | Self as a source of success/failure | Child's influence on/control over the outcomes of their own behaviors. | SELFSUC-CESS |
| | | Independence | Child's self-reliance and independence. This includes child's independent interactions with the physical environment. | INDEP |
| | | Self-direction | Child's self-directed, self-controlled behavior, involving creation, experimentation, and attainment of individual goals. | SELF-DIR |
| | | Resilience | Child's capacity to cope with difficulty, challenges and stress (e.g., self-soothing). | RESIL |
| **The uncontrollable matters for lifetime success** | | Career and income | Explicit mentions of career and income. | CAREER |
| | | Risks and uncertainty | Mentions of risk, uncertainty, surprise in relation to individual decisions and actions. | RISK |
| | | Lack of agency | Mentions of winning and losing in games, or other events where child's agency is limited or none. | LACK |

*(Continued)*

**Table 2.** (Continued)

| Theme | Sub-theme | Indicator | Definition | Abbreviation |
|---|---|---|---|---|
| **Individuals critically depend on others for their lifetime success** | *Individuals critically depend on teachers for lifetime success* | Guidance | Child's behavior through which s/he seeks guidance, support, and modelling, OR adult behavior through which guidance, support and modelling are offered to the child. | TEACH_GUIDE |
| | | Care | Child's behavior through which s/he seeks physical and/or emotional care, OR adult behavior through which physical and/or emotional care are offered to the child. | TEACH_CARE |
| | | Inclusion | Child's and others' activities aimed at including everyone, regardless of their physical attributes, background etc. | TEACH_INCL |
| | *Individuals critically depend on peers, family and the community for lifetime success* | Care | Child's behavior through which s/he seeks physical and/or emotional care, OR adult behavior through which physical and/or emotional care are offered to the child. | OTHER_CARE |
| | | Collaboration and teamwork | Child's engagement in collaboration or cooperation with others, typically to achieve a common goal. Also child's active involvement in groups or teams. | OTHER_COLLAB |
| | | Community | Child's reliance on and knowledge of others in the community, their roles, and/or contribution to the community life. | OTHER_COMMUNITY |
| | | Solidarity | Child's readiness to help others in need and awareness of dependence on others for such help. | OTHER_SOLID |
| | | Inclusion | Child's and others' activities aimed at including everyone, regardless of their physical attributes, background etc. | OTHER_INCL |

familiarization, [61]). After the first read, the indicators were defined and coded for each relevant statement or paragraph by Rater 1 (Phase 4: coding and reviewing themes, [61]). In the final analysis, single occurrences of the indicators were summed up and converted into the percentage of all occurrences for all indicators in each document. This was done to account for diverse document formats. Thereafter, reviewing the themes (Phase 5) ensued to validate the indicators identified and coded by Rater 1. Eleven countries were randomly selected for independent coding. To ensure her own consistency in coding, Rater 1 coded the eleven countries again at this point. All indicators for the eleven countries were coded by Rater 1 (twice) and Rater 2, and the first thirteen indicators (skills and uncontrollable factors) were coded also by Rater 3 to establish whether the given ratings would sufficiently generalize. The instances of coding for each indicator were summed up for each Rater and Spearman's rho was calculated to establish the interrater agreement. Spearman's rho was used because the cross-country analysis relied on the overall numbers of indicator occurrences on a document, not statement/paragraph, level. Definitions for three pairs of indicators were deemed interchangeable (confidence and self-efficacy, persistence and perseverance, collaboration and teamwork); these indicators were merged within pairs (confidence/self-efficacy, persistence/perseverance, collaboration/teamwork).

Subsequently, the indicators with the non-significant inter-rater agreement were dropped (Spearman's rho, see S1 Table in Supporting Material 1), resulting in a pool of thirty indicators. In the next step, correlations between the remaining indicators were calculated to estimate whether the indicators were sufficiently separable (Pearson's *r*, see S2 Table in Supporting Material 1). Five correlations greater than or equal to 0.6 occurred between (1) socioemotional skills and self-regulatory skills, (2) career-income and risk-uncertainty, (3) confidence/self-efficacy and resilience, (4) independence and autonomy, and (5) autonomy and self-direction. As socioemotional and self-regulatory skills pertained to the same theme, and the definition of self-regulatory skills overlapped with both socioemotional and cognitive skills, the indicator of self-regulatory skills was dropped. Career-income and risk-uncertainty occurred rarely, and typically in the same curricula, which likely drove the correlation; for this reason, both indicators were retained. This was also the case for confidence/self-efficacy and resilience. As the definition of autonomy was close to the definitions of independence and self-direction,

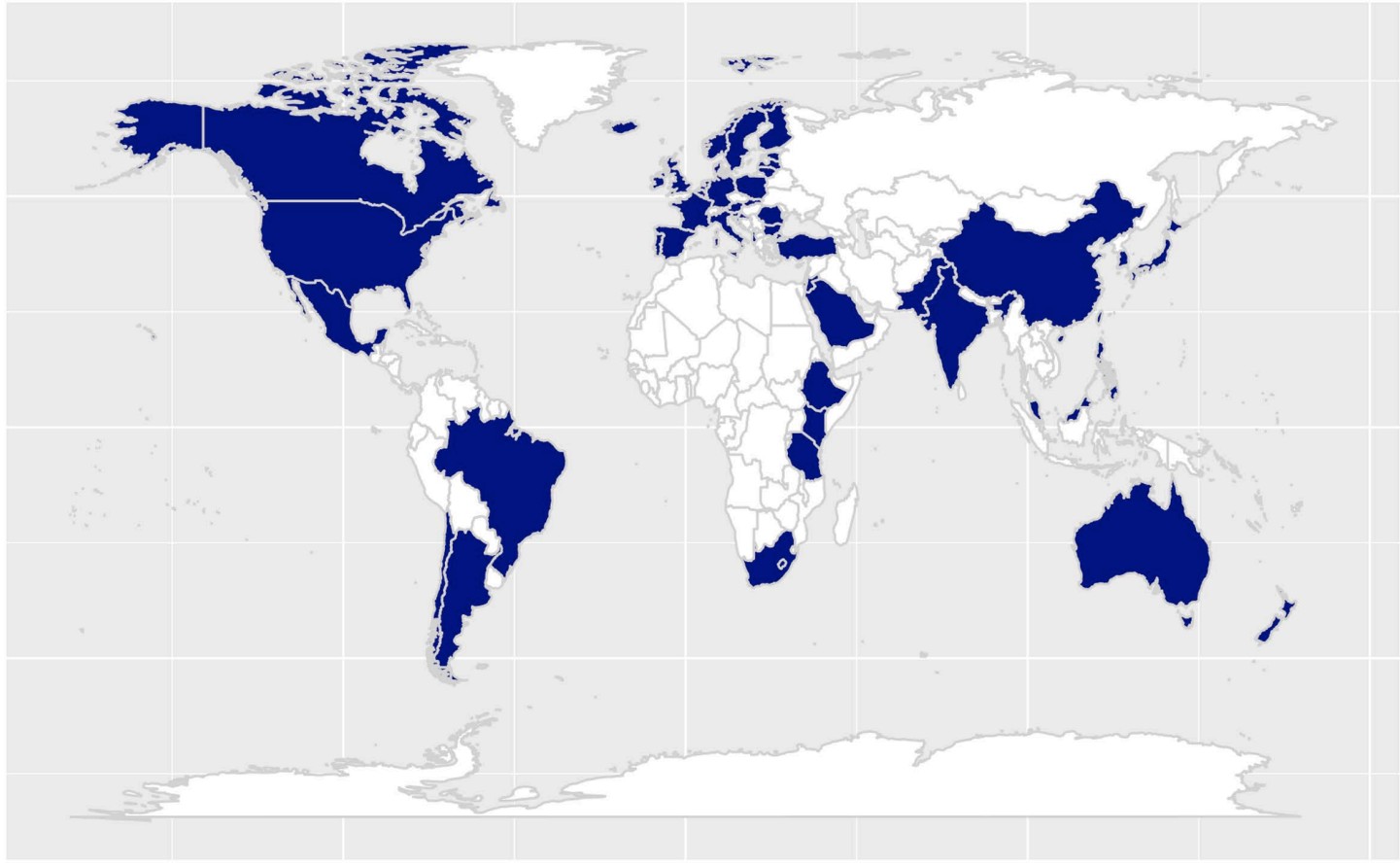

**Fig 1. Overview of the fifty-four countries considered in this study (in blue).** This figure was generated with R, using the following packages: ggplot2, rnaturalearth, rnaturalearthdata, cowplot, googleaway, ggrepel, ggspatial, libwgeom, sf, dplyr.

this indicator was dropped. Overall, twenty-eight indicators were included in the final analysis. Full results are available in S1 File (supporting dataset).

The raw data contained the number of occurrences for each indicator in each document. In the processed data, the number of occurrences of each indicator was divided by the sum of occurrences of all indicators in each document to obtain the relative indicator frequency. Some documents contained far more statements than others, allowing for standardising the data ahead of cross-organisational and cross-country comparisons. To enable such comparisons, the processed data was averaged across the documents available for each organization/country. For instance, if only one document was available, the frequency of occurrence of Indicator 1 in that document was simply retained; if two documents were available, an average of the frequencies of Indicator 1 in these two documents was calculated, and so on. The final dataset (S1 File) contained, therefore, a single relative indicator frequency per an organisation/country.

## Results

### (H1) Salience attributed to different types of skills

The close reading of the policymakers' reports (see Table 2) confirmed that experts and policymakers widely recognized the importance of universal and equal access to ECE [1,2] and encouraged countries worldwide to invest in access and

quality of ECE [2] to foster future economic returns and improve children's health, educational attainment, and social cohesion [1,2]. In most documents analyzed, outcomes related to social cohesion were typically estimated via economic indicators, such as decreased public expenditure on criminal justice and supporting victims of crime [[1]; Annex D]. Additionally, while the data showed that individual gains were typically discussed in relation to health, well-being, and cohesion, societal gains were typically reported in economic terms [2]. There were, however, some recent international reports concerned with early citizenship skills, highlighting the need of fostering participation, initiative, compassion, mutual respect and appreciation for equality and diversity [146]. For an overview of the themes and indicators per country, see Fig 2.

A total of six components of individual skillset was included in the thematic analysis and showed a sufficiently strong interrater agreement: (1) cognitive, (2) socioemotional, (3) physical, (4) self-care, (5) art, (6) citizenship skills. In the same category, achievement, understood as "explicitly mentioned individual achievement, goals or accomplishment", was also included. For skills' definitions and details of interrater agreement, see Tables 2 and S1 Table in Supporting Material 1. A considerable chunk of the content focused on skills in both international reports (*M* = 58.99%, *SD* = 13.93%*),* and government-endorsed curricula (*M* = 47.84%, *SD* = 14.74%). Both document types focused predominantly on cognitive skills (international reports: *M* = 23.01%*, SD* = 0.99%; government-endorsed curricula: *M* = 29.77%, *SD* = 9.99%*).* Interestingly, while socioemotional skills were a prominent component of international reports *(M* = 26.18%*, SD* = 7.27%*),* this was

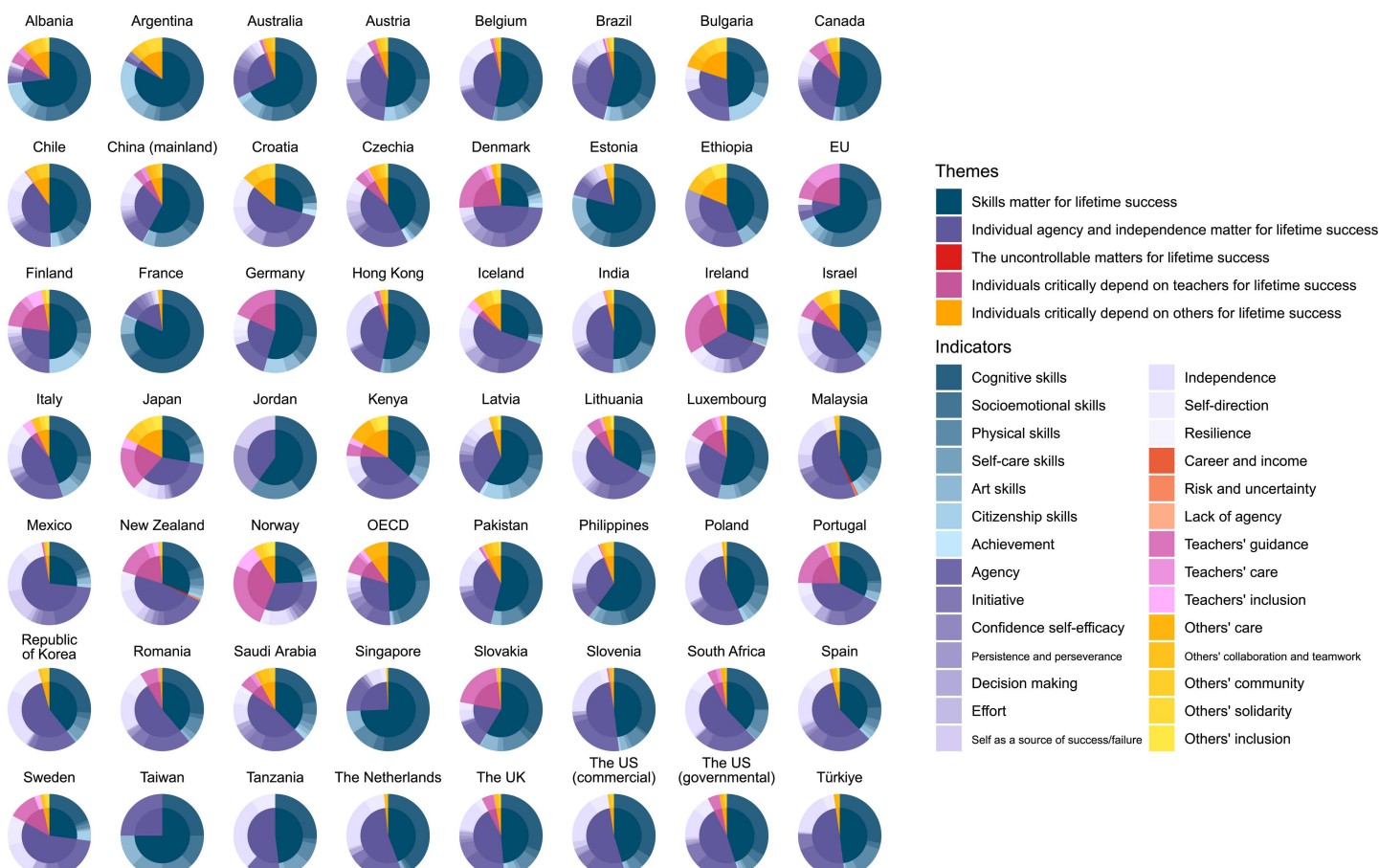

**Fig 2. Representation of themes and indicators in international reports (EU and OECD) and government-endorsed curricula.**

typically not the case for the government-endorsed curricula *(M* =4.01%, *SD* =2.92%). This reveals an overwhelming focus on individuals' cognitive skills, followed by socioemotional skills, in defining life success, compared to other skills such as citizenship skills.

Globally, these results are consistent with H1, suggesting that education stakeholders prioritize children's skills linked to individual lifetime success and associated with their market value, focusing particularly on cognitive skills and –mostly within international organizations – on socioemotional skills, whereas much less relevance was given to social skills linked to solidarity and cooperation, such as citizenship skills.

**(H2) Agency and independence versus uncontrollable factors and role of others in explaining individual lifetime success**

Ten aspects of individual agency and independence identified in the analysis showed a sufficiently strong interrater agreement: (1) agency, (2) initiative, (3) confidence and self-efficacy, (4) persistence and perseverance, (5) decision making, (6) effort, (7) self as a source of success/failure, (8) independence, (9) self-direction, and (10) resilience. These aspects were strongly emphasized in the government-endorsed curricula (*M* =40.50%, *SD* =13.44%), and, to a lesser extent, in the international reports (*M* =19.66%, *SD* =15%). The emphasis in the government-endorsed curricula was mostly linked to individual agency (*M* =15.33%, *SD* =5.35%)

Three further aspects that may contribute to lifetime success were identified, but these were rarely present in the international reports and government-endorsed curricula: (1) career and income, (2) risks and uncertainty, and (3) lack of agency. Note that we defined uncontrollable external events as society-wide political and economic contexts, the monetary value of different careers, and events where individual agency is limited or none. These aspects were largely absent in the international reports, being present only in 4 out of 53 government-endorsed curricula, taking up a minuscule portion of the content (*M* =0.59%, *SD* =0.46%). The countries that did mention uncontrollable factors were Ireland, Malaysia, New Zealand, and the Philippines (see Fig 2).

In the analysis, teacher support was coded separately from support received from others (e.g., peers, family, and members of the community). Dependence on teachers was emphasized in international reports (*M* =16.38%, *SD* =8.1%), but it was negligible in the government-endorsed curricula (*M* =6.16%, *SD* =8.47%). Dependence on others was rarely mentioned in both document types (international reports: *M* =4.97%, *SD* =7.03%; government-endorsed curricula: *M* =5.45%, *SD* =4.92%). Three aspects of teacher support showed a sufficiently strong interrater agreement: (1) guidance, (2) care, (3) and inclusion, and this was the case for five aspects of others' support: (1) care, (2) collaboration and teamwork, (3) community, (4) solidarity, and (5) inclusion. The aspects relevant for appreciating and fostering social cohesion, such as collaboration and teamwork, community, solidarity, and inclusion were rather underrepresented in both international reports (collaboration and teamwork: *M* =0%, *SD* =0%, community: *M* =0.27%, *SD* =0.38%, solidarity: *M* =0%, *SD* =0%, teachers' inclusion: *M* =0.93%, *SD* =1.31%, others' inclusion: *M* =0%, *SD* =0%) and government-endorsed curricula (collaboration and teamwork: *M* =1.61%, *SD* =1.6%, community: *M* =1.62%, *SD* =1.71%, solidarity: *M* =0.63%, *SD* =1.32%, teachers' inclusion: *M* =0.85%, *SD* =1.79%, others' inclusion: *M* =0.82%, *SD* =1.37%).

Overall, these results aligned with H2 stating that education stakeholders emphasize children's agency and independence in lifetime success, while minimizing the key role of uncontrollable factors and others' actions (i.e., their social and community context) in contributing to individual achievement.

## Discussion

This study is –to our knowledge– the first to examine whether key education stakeholders are globally promoting an economic meritocratic mindset in ECE. To do so, we gathered and analyzed rich narrative data from advisory reports of international organizations and from reformed early childhood curricula of 53 countries across six continents from 1999 to 2023.

First, we found that, while experts and policymakers recognized the importance of ECE access and quality, they defined social cohesion primarily based on economic indicators. Second, in line with Hypothesis 1, the ECE policy documents did prioritize cognitive skills and –mostly among international organizations– socioemotional skills as key for individual achievement, but citizenship skills were largely omitted. Third, findings suggested that individual agency and responsibility were defined as the most important factors behind individual achievement in education and the labor market in ECE reforms by international organizations and governments. By contrast, factors that are rather "uncontrollable" (e.g., intergenerational transmission of the socioeconomic advantage and disadvantage, good or bad fortune, economic demand on individual skills, economic valorization of one's own profession, and the support of other members of the community and the society) received much less attention. These findings support Hypothesis 2. These results specifically reveal that international organizations and governments strongly embraced an economic meritocratic mindset in their ECE, observed in the notion that life outcomes mainly depend on talent and effort. Meanwhile, the idea that individuals' life trajectories crucially depend on external support from peers, family members, communities or institutions was largely missing in the analyzed ECE documents.

The extensive thematic analysis carried out in the present study suggests that policymakers, both at the international and at the country level, implicitly share the economic meritocratic mindset. These findings show, in line with previous findings on neoliberalism in educational policy [42,44,46], that children's skill, agency, and independence have become an integral part of ECE curricula. This suggests that the economic meritocratic mindset might have been planted in the curricular reforms worldwide. This mindset emphasizes individual agency, skills, and responsibility for one's own academic and socioeconomic achievement, mirroring the child-centered discourse that views the child as a self-governing, responsible individual capable of initiative, persistence, and autonomous decision making [46]. We found relatively few mentions of the social interdependence for lifetime success in the advisory reports, while three-fourths of the governmental documents provided such mentions. The mentions of the factors beyond individual's control were even scarcer. Thus, these findings suggest, in line with our hypotheses, that ECE agendas are aligned with the economic meritocracy mindset neglecting the influence of uncontrollable factors on individual achievement, and promoting individual agency in own achievement with economic-based returns, while often disregarding the critical contribution of others in shaping individuals' life trajectories.

Our findings situate the ECE reforms in the context of meritocratic beliefs rather than neoliberalism, contrary to previous ECE research. We recognize that neoliberalism heavily draws on and promotes the economic meritocratic mindset. However, we narrowed our focus to this mindset specifically, due to its intrinsic role in contemporary educational systems. Our findings align with some previous research, which likewise found individualistic mindset across policy documents [38]. The promotion of individual, cognitive or socioemotional, skills, individual agency, and responsibility is fully aligned with the vision of a self-governing child, responsible for own successes and failures. Our results, therefore, corroborate a vision of child's autonomy and responsibility as an individual trait [42,46], showing how contemporary international ECE institutions promote individual responsibility far more than social responsibility and social cooperation.

Our study had some limitations that should be addressed in future studies. First, while this study presented detailed evidence of the discourses and proposals shaping ECE agenda at the international and the country level, we have not zoomed in on community and local contexts which may significantly shape teaching and learning activities in the classroom. Critically, research on published documents and interviews with various school-level educational stakeholders, such as principals and educators, and how ECE programs are internalized by families and students, is essential to understand the specific skills that ECE instill in practice in everyday life, irrespective of national-level discourses and recommendations.

Second, for reasons of space, our study has not paid much attention to examine cross-country and cross-cultural variations in how ECE documents link to the meritocratic economic mindset. Although our analyses did suggest a certain convergence, future studies, ideally combining different qualitative and quantitative techniques, should address the role of country-level institutional factors in shaping ECE government-endorsed documents. The present study focused

on similarities rather than differences between the states in promoting the economic meritocratic mindset, but country or regional differences should receive further attention. For instance, previous studies showed that, even though several policy documents may invoke the vision of children as active constructors of own knowledge about the world, this may involve diverse terms, such as "purposeful play" in the Singaporean context, "play-based learning" in the Chinese context, and "learning through play" in the Australian context [31]. Such regional differences may be key when uncovering the traces of the economic meritocratic mindset across country-specific policies.

Third, our study lacked a focus on trends over time. Future studies would benefit from a large-scale, historical approach, previously presented for the primary educational context between 1965 and 2000 [24], which could elucidate how the ECE reforms changed over time in each country and region. This would certainly better address – support or refute – the argument that the international reports issued by the World Bank, the EU or the OECD were instrumental in promoting the economic meritocratic mindset in the country-specific ECE policies.

Despite some limitations, our study has shown that ECE reforms worldwide put an emphasis on skills and individual agency as the means to society-wide readiness for the 21st century challenges. While skills support individuals and societies in dealing with the ongoing social and technological changes, this is not all that it takes to foster social cohesion. If the ECE system instills in individuals a belief that talent and skill determine their self-worth, and that their achievement (or lack thereof) is their sole responsibility, this may reduce individuals' understanding of how other factors influence achievement. As a result, individuals may lack tools to understand that uncontrollable factors (e.g., their social environment) critically influence their own and others' life course. This does not take away the importance of individual agency and responsibility for own achievement. Yet, it paints a more complex picture that can potentially equip current and future citizens with better abilities to build more inclusive and cooperative societies. While academic and socioemotional skills are present in the reformed curricula, a reflection on how these skills are framed and utilized in the 21st century socio-economic systems is largely missing. This is quite surprising, considering the well-documented role that socioeconomic systems and structures shape individuals' present and future opportunities [150].

Our findings have real-world implications for how countries articulate their rationale for ECE investment, yet it is important to consider that different actors and stakeholders (e.g., communities versus policy makers) may have different views and priorities in relation to ECE. First, as Patnaik [151] demonstrates in the Indian context, there may be a marketing mismatch between the meritocratic framing of ECE and the cultural values or expectations of families. When governments promote ECE as a tool for economic growth or individual success, these messages may not resonate with communities that prioritize other developmental goals, potentially limiting engagement. Second, framing ECE as a high-return investment can contribute to the emergence of a competitive marketplace, particularly in highly privatized educational settings (e.g., the Chilean context; [152]), where families begin to seek out the "best" ECE programs to get ahead—much like they do with elite universities. Finally, these ECE dynamics can exert pressure on ECE teachers who emphasize citizenship and equity in their professional ethos and values (e.g., [153]). Therefore, parents influenced by meritocratic framings may expect educators to focus on hard cognitive skills and measurable competitive academic success instead. As a result, the meritocratic logic may be reinforced, potentially reshaping classroom practices in ways that educators may not necessarily support. Furthermore, neoliberal ECE policies may foster teacher's resistance (e.g., in Australia: [154]; South Africa: [155]; the United Kingdom: [156]; the United States: [157,158]; see also [159,160]), and so future research should dive into the dynamics of the relationship between different actors and stakeholders, an issue that, due to the study's focus, we have not addressed here.

Our study has direct scientific and policy implications. In the last two decades, states have readily invested in individual citizens to ensure future economic prosperity [1,16,53,55,57]. That the peace and prosperity of future societies hinge on social cohesion has been recognized in several international reports on early childhood education [e.g., 5,15,16]. This has, however, not become a prominent theme across the reformed ECE curricula. Reinforcing children's sense of agency in their own achievement, without encouraging an explicit awareness that other members of the society have actively

contributed to such achievement, contributes to weaker social cohesion within societies. Given that economic meritocracy strains individual health and psychological well-being across social strata [8], its impact on ECE curricula demands prompt attention. Future ECE reforms may foster an early understanding that external events, be it lucky or unlucky ones, and the support of others, critically shape individual achievement. As long as ECE reforms imply that individual success does not depend on contributions from others, countries will remain unprepared to collectively respond to the most pressing current societal challenges. We encourage researchers in academia and beyond to further investigate the role of ECE reforms. Scholars could, for instance, consider the role of multiple actors and stakeholders in shaping societal values, emphasizing the processes that operate in the classroom, and applying diverse methodologies (e.g., longitudinal methods, mixed methods) across different ECE reforms and policy contexts.

## Supporting information

**Supporting Material 1.  (separate file). Supplementary Tables S1-S3.**
(DOCX)

**S1 File.  (separate file). An overview of a composition of policy decrees and ECE curricula across the OECD, the EU, and the fifty-four countries included in the study.**
(XLSX)

## Acknowledgments

We thank Cassandra Origer for her help with interrater agreement and presentation of the data in Table 2. Furthermore, we thank Ryszard Bobrowicz for his help with data visualization, particularly Fig 2.

## Author contributions

**Conceptualization:** Katarzyna Bobrowicz, Pablo Gracia, Samuel Greiff.

**Data curation:** Katarzyna Bobrowicz.

**Formal analysis:** Katarzyna Bobrowicz, Ziwen Teuber.

**Investigation:** Katarzyna Bobrowicz.

**Methodology:** Katarzyna Bobrowicz, Pablo Gracia.

**Project administration:** Katarzyna Bobrowicz.

**Validation:** Katarzyna Bobrowicz.

**Visualization:** Katarzyna Bobrowicz.

**Writing – original draft:** Katarzyna Bobrowicz, Pablo Gracia.

**Writing – review & editing:** Katarzyna Bobrowicz, Pablo Gracia, Ziwen Teuber, Samuel Greiff.

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
