## [Decision Letter · Decision Letter 0]

1 Sep 2024

Dear Dr. Bobrowicz,

Thank you for submitting your manuscript to PLOS ONE. After careful consideration, we feel that it has merit but does not fully meet PLOS ONE’s publication criteria as it currently stands. Therefore, we invite you to submit a revised version of the manuscript that addresses the points raised during the review process.

We look forward to receiving your revised manuscript.

Kind regards,

Daphne Nicolitsas

Academic Editor

PLOS ONE

2. We note that Figure 1 in your submission contain [map/satellite] images which may be copyrighted. All PLOS content is published under the Creative Commons Attribution License (CC BY 4.0), which means that the manuscript, images, and Supporting Information files will be freely available online, and any third party is permitted to access, download, copy, distribute, and use these materials in any way, even commercially, with proper attribution. For these reasons, we cannot publish previously copyrighted maps or satellite images created using proprietary data, such as Google software (Google Maps, Street View, and Earth). For more information, see our copyright guidelines: http://journals.plos.org/plosone/s/licenses-and-copyright.

Reviewers' comments:

Reviewer's Responses to Questions

**Comments to the Author**

1. Is the manuscript technically sound, and do the data support the conclusions?

Reviewer #1: Partly

2. Has the statistical analysis been performed appropriately and rigorously?

Reviewer #1: No

3. Have the authors made all data underlying the findings in their manuscript fully available?

Reviewer #1: Yes

4. Is the manuscript presented in an intelligible fashion and written in standard English?

Reviewer #1: Yes

Reviewer #1: Dear Authors,

Thank you for the opportunity to review your manuscript. I commend your efforts in exploring whether reformed Early Childhood Education (ECE) can lay the foundation for an economic meritocratic mindset across fifty-four countries. The results of your thematic analyses are both intriguing and potentially impactful in the fields of ECE and related educational disciplines, particularly regarding the development of policies and reforms that consider a meritocratic mindset. In the document attached to this review, I provide detailed comments and suggestions that I believe will enhance the clarity, structure, and overall contribution of your work.

Best wishes,

Reviewer

**Do you want your identity to be public for this peer review?** For information about this choice, including consent withdrawal, please see our Privacy Policy

Reviewer #1: No

---

## [Author Response · Author response to Decision Letter 1]

19 Nov 2024

Dear Editorial team and Reviewer 1,

We are glad that the Editorial team and the anonymous reviewer see merits in our paper considered for publication at PLOS ONE. Reviewer 1 has provided excellent suggestions and comments that we have carefully considered to improve our manuscript. Below, we provide a specific answer to each point raised by Reviewer 1. We have carefully answered each point and accordingly made important improvements to the and clarified any issues in this response letter. To ease transparency and clarity in the reviewing process, the changes introduced to the manuscript are now highlighted in yellow throughout the manuscript.

In our own reading of the manuscript, we have noticed that the analysis involved fifty-three, not fifty-four countries, as the commercial and the governmental curricula from the United States of America were considered as coming from two different countries. We have now amended this throughout the manuscript. Furthermore, we have amended Fig 2 to include data from France among the pie charts, which has been omitted in the original version of the figure. Finally, we have amended the title to better reflect the message of the manuscript and its relationship to a key source (“The Meritocracy Trap” by D. Markovits).

We are looking forward to further feedback from the Editorial team and the Reviewer in the hope that the revised version of the manuscript is meeting the publication standards of PLOS ONE.

Best regards,

The Authors

Response to the Editor

In the revised submission, we have followed the PLOS ONE formatting guidelines and the templates for file naming.

2. We note that Figure 1 in your submission contain [map/satellite] images which may be copyrighted. All PLOS content is published under the Creative Commons Attribution License (CC BY 4.0), which means that the manuscript, images, and Supporting Information files will be freely available online, and any third party is permitted to access, download, copy, distribute, and use these materials in any way, even commercially, with proper attribution. For these reasons, we cannot publish previously copyrighted maps or satellite images created using proprietary data, such as Google software (Google Maps, Street View, and Earth). For more information, see our copyright guidelines: http://journals.plos.org/plosone/s/licenses-and-copyright.

Many thanks for drawing our attention to this matter. The map was generated through a standard “ggplot” function relying on the following packages in R: “ggplot2”, “ggpubr”, “rnaturalearth”, “rnaturalearthdata”, “cowplot”, “googleaway”, “ggrepel”, “ggspatial”, “libwgeom”, “sf”. Using R for this purpose has now been clarified in legend of Fig 1 (page 41). We have not cited all these packages to save space in the current revised version.

The following code was used to generate Fig 1:

“require(ggplot2)

require("rnaturalearth")

require("rnaturalearthdata")

require("cowplot")

require("googleaway")

require("ggrepel")

require("ggspatial")

require("libwgeom")

require("sf")

world <- ne_countries(scale = "medium", returnclass = "sf")

class(world)

ggplot(data = world) +

geom_sf()

ggplot(data = world) +

geom_sf(color = "darkgray", fill = "white")

map<-map_data("world")

require(dplyr)

# Set colors

map <- mutate(map, fill = ifelse(region %in% c("Ethiopia",

"Kenya",

"South Africa",

"Tanzania",

"China",

"Hong Kong",

"India",

"Japan",

"Malaysia",

"Pakistan",

"Philippines",

"South Korea",

"Singapore",

"Taiwan",

"Australia",

"New Zealand",

"Albania",

"Austria",

"Belgium",

"Bulgaria",

"Croatia",

"Czechia",

"Denmark",

"Estonia",

"Finland",

"France",

"Germany",

"Iceland",

"Ireland",

"Italy",

"Latvia",

"Lithuania",

"Luxembourg",

"Norway",

"Poland",

"Portugal",

"Romania",

"Slovakia",

"Slovenia",

"Spain",

"Sweden",

"Netherlands",

"UK",

"Jordan",

"Israel",

"Saudi Arabia",

"Turkey",

"Canada",

"Mexico",

"USA",

"Argentina",

"Brazil",

"Chile"), "navy", "white"))

# Use scale_fiil_identity to set correct colors

ggplot(map, aes(long, lat, fill = fill, group=group)) +

geom_polygon(colour="lightgray")+

scale_fill_identity()+

theme(axis.title = element_blank(),

axis.ticks = element_blank(),

axis.text =element_blank())”

Response to the Reviewer

Thank you for the opportunity to review this manuscript. I commend your e�orts in exploring whether reformed Early Childhood Education (ECE) may lay the foundation for an economic meritocratic mindset across �fty-four countries. The results of your thematic analyses are both intriguing and potentially impactful in the �elds of ECE and related educational disciplines, particularly in the context of developing policies and reforms that consider a meritocratic mindset. Below, I provide detailed comments and suggestions that I believe will enhance the clarity, structure, and overall contribution of your work:

We want to thank Reviewer 1 for providing such detailed and constructive review which has been instrumental to revising and improving our manuscript.

1. The introduction e�ectively builds the arguments necessary to examine the hypotheses. However, while the authors state that the role of meritocratic ideologies within the context of ECE has been overlooked in previous research, I did not �nd supporting evidence for this claim. To strengthen the manuscript, I suggest either presenting related arguments or citing earlier studies that highlight this gap in the literature. Additionally, it is important to consider that readers are interested in understanding the rationale for conducting the research, not simply because it �lls a gap in the literature. If this research area is indeed underexplored, it would be more compelling to build arguments around the signi�cance and potential impact of studying this topic, rather than merely stating that it is a gap. There are many unexplored phenomena in the literature, but the key question is: why is it important to investigate this particular phenomenon?

We thank Reviewer 1 for this comment. We fully agree with Reviewer 1 that the previous version of the paper required more direct engagement with previous scholarship to motivate the topic and our study in the context of existing literature. In the introduction of the new version of the paper we have now:

a) contextualized the present study in relation to previous research on neoliberalism, the Human Capital Theory, and meritocracy in educational policy in ECE and beyond (pages 5-6). We have highlighted the gaps in the literature, while also situating previous literature and previous arguments and research more in depth (pages 4 and 6).

b) stressed more directly the motivation of this topic from the beginning in the introduction of the manuscript (page 3).

Finally, in line with the comments from Reviewer 1, we have also revised the abstract to communicate more effectively the motivation, objectives, methodology, results and overall conclusion of our study (page 2).

2. The authors state, “The hypothesis may be rejected if education stakeholders indicate that other factors, such as the current market value of one’s skills and uncontrollable external events, can critically in�uence one’s lifetime success” (p. 6). In research, it is important to provide supporting arguments for these factors. I noticed that the study does not include a Literature Review section, which could be a suitable place to present these factors in detail. While the current market value of skills and uncontrollable external events are mentioned, the manuscript does not provide a clear �ow of ideas to guide readers in recognizing the signi�cance of these potential factors. Including a Literature Review or integrating more supporting arguments could improve the coherence and depth of the discussion.

We agree that the background and the literature review were underdeveloped in the original version of the manuscript. In the revised version, we have (1) added a Literature Review section in the Background, (2) revised and improved the theoretical framework building on previous research in this area more explicitly, and (3) clarified more thoroughly what is meant by the current market value of skills and uncontrollable external events (pages 4 and 7).

3. I am not entirely certain whether PLOS ONE requires a section on Theoretical, Conceptual, and/or Analytical Frameworks. However, I strongly recommend integrating these frameworks into your manuscript, as their omission could undermine the credibility of the results. Speci�cally, it is challenging to accept the results presented in the study without a detailed explanation of its foundational elements, such as the Theoretical and Analytical Frameworks, especially given the use of thematic analysis. An Analytical Framework, for instance, would provide valuable insight into the origin of the trends, themes, and patterns identi�ed by the authors. Including these frameworks is not only essential for the rigor of the research but also bene�cial for other researchers who may wish to replicate the study in di�erent contexts.

We fully agree with Reviewer 1 in that adding an Analytical Framework provides valuable insight into the origin of the trends, themes, and patterns identi�ed”. As mentioned in point 2, we have extensively added a new section for these purposes and then improved our Background, in terms of literature review and theoretical expectations (pages 5-6). Our methodological approach is spelled out and inserted right after the Background in the revised manuscript (pages 8-12).

4. The authors state, “We de�ned uncontrollable external events as society-wide political and economic contexts, the monetary value of di�erent careers, and events where individual agency is limited or none” (p. 7). While I can accept these events as variables, I would like to see examples of how such events are incorporated into the research questions (hypotheses). The absence of a Literature Review makes it challenging to justify their inclusion in the study. It is easy to identify variables that seem relevant and interesting, but justifying why these particular variables should be part of the phenomenon under investigation requires robust arguments (rationale). Consider this question: why should these speci�c variables (as highlighted above) be selected over others? I am not trying to make this process di�icult for the authors, but as a reader, I would appreciate seeing how these variables are integrated into the research. Please re�ect on this question when revisiting the hypotheses.

We fully agree with Reviewer 1 in that justifying this conceptualization based on previous literature and on an argument that derives from it is important for the study. Therefore, the background has been changed globally following these recommendations from Reviewer 1, including an explanation of “uncontrollable factors” in the Introduction (page 4).

5. The Analytical Framework guides the selection of the analysis method, and in this research, the authors employed thematic analysis. However, it is unclear how the authors conducted this thematic analysis. Speci�cally, how were the themes in Table 2 identi�ed? What was the rationale for retaining these particular themes? Additionally, how many coders were involved in the analysis, and how were discrepancies between the coders’ results resolved? These important details should be clearly outlined in a Method section, which is currently missing from the manuscript. Including this information would enhance the transparency and reliability of the research process, thereby strengthening the �delity of the study's �ndings. Furthermore, since the study leans towards a qualitative approach, it is important to explain how triangulation was addressed.

We appreciate this remark, and we have striven to clarify the entire methodological process behind the analysis. In line with the Reviewer’s feedback and the PLOS One guidelines, we have now included a “Materials and methods” section between the Background and the Results. done it. Critically, in the revised manuscript, we have extensively revisited the method by accounting for the Reviewer’s questions on the flow of the thematic analysis (pages 9-11), the definition of the themes and the rationale behind them (page 10), and the combination of the qualitative and the quantitative analyses (page 12).

6. The authors state, “Themes and indicators of a meritocratic mindset were identi�ed based on previous literature and close, iterative reading of the data (for details consult Materials and Methods)” (p. 14). However, I could not locate a Materials and Methods section. Toward the end of the article, I found supplementary materials, which I assume is what the authors referred to. My suggestion is to incorporate the methodological analysis into a dedicated Method section within the manuscript. Additionally, the rationale for selecting Spearman’s rho to �nd correlations among ratings of curricula should be clearly justi�ed. Why was this method chosen instead of inter-rater correlations, given that the raters conducted the thematic analysis? If the authors followed a di�erent analytical process, it should be clearly explained in the manuscript. For instance, it is unclear how the data in the Excel were generated, as there is no description or explanation provided. The manuscript primarily mentions that thematic analysis was conducted, but the details of this process need to be thoroughly articul

---

## [Decision Letter · Decision Letter 1]

15 Apr 2025

Dear Dr. Bobrowicz,

Thank you for submitting your manuscript to PLOS ONE. The reviewer acknowledges that the revisions asked in the previous round have been performed. As a new reviewer they ask for an additional minor revision which I think can be easily met. Therefore, we invite you to submit a revised version of the manuscript that addresses the points raised during the review process.

We look forward to receiving your revised manuscript.

Kind regards,

Daphne Nicolitsas

Academic Editor

PLOS ONE

Journal Requirements:

Reviewers' comments:

Reviewer's Responses to Questions

**Comments to the Author**

Reviewer #2: All comments have been addressed

2. Is the manuscript technically sound, and do the data support the conclusions?

Reviewer #2: Yes

3. Has the statistical analysis been performed appropriately and rigorously?

Reviewer #2: N/A

4. Have the authors made all data underlying the findings in their manuscript fully available?

Reviewer #2: Yes

5. Is the manuscript presented in an intelligible fashion and written in standard English?

Reviewer #2: Yes

Reviewer #2: This is a very interesting paper, and the revisions are important and substantive. Although I was not a first-round reviewer, the authors have engaged substantively with the comments and it has made for a much improved paper. are two real issues that I would suggest for minor revision. First, there is a seminal paper written by Careiro & Heckman from 2003 [Carneiro, P. M., & Heckman, J. J. (2003). Human capital policy.].

After this paper, Heckman used his significant power and advantage in the field of economics to explicitly promote ECE as "the best ROI". This not only turned ECE into an "economic investment in meritocracy" but also specifically set out via intermediary orgs, international NGOs and policy entrepreneurs to invest the world in ECE as prevention of bad outcomes like public dependency or crime (like public health) and the production of good outcomes (executive functioning, emotional regulation, and human capital more broadly). First, this paper should definitely be included in your lit review as should many of those that follow & cite this paper. It also may need to cite some of the foci of the programs that were included in Heckman's original analyses and their foci -- Head Start, for example, had a very large focus on citizenship, community advancement, and equity. So here, it's worth noting what the human capital focus by Careiro & Heckman lifts to the top & leaves behind.

However, the inclusion of this paper and its role in the proliferation of the meritocratic promises of ECE present a few challenges & opportunities that this history presents for your findings as currently written:

1) the spread of meritocratic ideas is currently unembodied in your paper. Yet, any study of diffusion & institutionalization would note that intermediary orgs, bodies of evidence, and institutional entrepreneurs are typically pushing particular ways of seeing and using evidence to achieve policy ends. Here's an example of this that followed Heckman's article within the year following its publication [Bruner, C. (2004). Many happy returns: Three economic models that make the case for school readiness. Des Moines, IA: State Early Childhood Policy Technical Assistance Network. Retrieved May, 31, 2005.].

2) if this is the case, then it's possible that the values written about are simply the articulations that neoliberal governments use to justify the policy of ECE...e.g. governments won't make the case for ECE on these other means and the only ones that do are actually demonstrating a measure of "looser ties" with the intermediary orgs that spread ECE policy. So, your measures might in fact demonstrate the level of influence of this kind of tight ties among the neoliberal countries but not actually reflect the reality of ECE programs or values on the ground. This is especially likely given that ECE & almost all K-3 education is notoriously localized and quite resistant to efforts to standardize due to a lack of common training & certification of teachers & low pay in many countries. So, you would want to clarify at what level you think the ideology is reflected (I don't think you're saying it's for sure happening on the ground, but you'll want to be very clear about your language to designate that this is the way countries are justifying their investments & priorities to external audiences).

3) the opportunity here is that in the implications/discussion, there are a few missing that really matter for kids and families as well as the take-up of ECE: a) Marketing mismatches may not appeal to families based on their cultural values ... that is, countries may articulate a meritocratic rationale and it may not resonate [see this example: Patnaik, N. (2023). Teaching development: Critically evaluating ECE for economic growth in India. International Journal of Social Science & Economic Research, 8(08), 2390-2396.], b) mismatches between the promises made for ECE and people's experiences may support a backlash or divestment in countries where there is a large uptake (i.e. "my ECE program is low-quality, didn't help my child", etc.)., c) the meritocratic promise doesn't only affect individual expectations, it creates a competitive market for ECE in places where it's privatized as families seek to "get ahead" via best ECE like best colleges, and d) ECE teachers who do focus on citizenship and/or social skills may experience pressure from families to "teach hard skills" for individual success (that is, parents as consumers may both believe the marketing & reinforce it's expectation on their ECE providers). Fleshing out these real-world implications of your research may help strengthen the rationale for the work you've done & demonstrate that a nation's articulated rationale for investing in ECE may actually, in the long-run, determine it's focus and content over time.

Best of luck as you continue to revise this paper. It's meaningful and important work, and I'd suggest that you follow-up with a paper on the proliferation of this model & its origins globally.

**Do you want your identity to be public for this peer review?** For information about this choice, including consent withdrawal, please see our Privacy Policy

Reviewer #2: **Yes: ** Bridgette Davis

---

## [Author Response · Author response to Decision Letter 2]

6 May 2025

Dear Editors and Reviewers,

Thank you very much for the opportunity to revise and resubmit the paper considered for publication in PLOS One.

We have carefully read and implemented the very relevant suggestions from Reviewer 2. Below we provide an answer to each comment raised by Reviewer 2 and reference to the action plan implemented in relation to these comments. We believe that our paper is now stronger and improved thanks to the feedback received through the hitherto revision rounds at PLOS One.

Best regards,

The Authors

RESPONSE TO REVIEWER 2

Reviewer #2:

This is a very interesting paper, and the revisions are important and substantive. Although I was not a first-round reviewer, the authors have engaged substantively with the comments and it has made for a much improved paper. There are two real issues that I would suggest for minor revision.

We are glad to hear that Reviewer 2 has found our paper very interesting and also values the extensive work and improvements made on the manuscript in the previous version of the paper.

First, there is a seminal paper written by Carneiro & Heckman from 2003 [Carneiro, P. M., & Heckman, J. J. (2003). Human capital policy]. After this paper, Heckman used his significant power and advantage in the field of economics to explicitly promote ECE as "the best ROI." This not only turned ECE into an "economic investment in meritocracy" but also specifically set out via intermediary orgs, international NGOs and policy entrepreneurs to invest the world in ECE as prevention of bad outcomes like public dependency or crime (like public health) and the production of good outcomes (executive functioning, emotional regulation, and human capital more broadly). First, this paper should definitely be included in your lit review, as should many of those that follow & cite this paper. It also may need to cite some of the foci of the programs that were included in Heckman's original analyses and their foci—Head Start, for example, had a very large focus on citizenship, community advancement, and equity. So here, it's worth noting what the human capital focus by Carneiro & Heckman lifts to the top & leaves behind.

We thank Reviewer 2 for this suggestion. We have now accordingly included the Carneiro and Heckman (2003) study in our literature review, along with a discussion of its outsized influence in shaping ECE as an economic investment, particularly through Heckman’s broader advocacy work. We also reference subsequent work that amplified this economic framing, including Bruner (2004), to highlight how these ideas circulated via policy networks and intermediaries. In addition, we have referred to programs like Head Start—emphasizing citizenship, community, and equity—in relation to this literature.

However, the inclusion of this paper and its role in the proliferation of the meritocratic promises of ECE present a few challenges & opportunities that this history presents for your findings as currently written:

1) The spread of meritocratic ideas is currently unembodied in your paper. Yet, any study of diffusion & institutionalization would note that intermediary orgs, bodies of evidence, and institutional entrepreneurs are typically pushing particular ways of seeing and using evidence to achieve policy ends. Here's an example of this that followed Heckman's article within the year following its publication [Bruner, C. (2004). Many happy returns: Three economic models that make the case for school readiness.].

We thank Reviewer 2 for drawing our attention to this matter. We have responded by explicitly incorporating the role of intermediary organizations, institutional entrepreneurs, and policy networks into our discussion of how ECE rationales are diffused and institutionalized. In particular, we refer to Bruner (2004) as a clear example of how economic models were quickly mobilized to support early learning investments. This allows us to better contextualize how the meritocratic framing of ECE has been promoted and sustained beyond the academic sphere.

2) If this is the case, then it's possible that the values written about are simply the articulations that neoliberal governments use to justify the policy of ECE... e.g., governments won't make the case for ECE on these other means and the only ones that do are actually demonstrating a measure of "looser ties" with the intermediary orgs that spread ECE policy. So, your measures might in fact demonstrate the level of influence of this kind of tight ties among the neoliberal countries but not actually reflect the reality of ECE programs or values on the ground. This is especially likely given that ECE & almost all K-3 education is notoriously localized and quite resistant to efforts to standardize due to a lack of common training & certification of teachers & low pay in many countries. So, you would want to clarify at what level you think the ideology is reflected (I don't think you're saying it's for sure happening on the ground, but you'll want to be very clear about your language to designate that this is the way countries are justifying their investments & priorities to external audiences).

We appreciate this comment and we agree with Reviewer 2. In the manuscript, we focus on what states, through their ECE policies, are aiming to promote in ECE facilities, acknowledging that such policies may cause resistance among teachers across countries, which may, however, dissipate with increased state support for implementing the policies. Therefore, in the revised version, we explicitly state that (1) educational stakeholders may hold diverse views, and (2) we propose that future studies should further investigate these dynamics across institutions and stakeholders.

3) The opportunity here is that in the implications/discussion, there are a few missing that really matter for kids and families as well as the take-up of ECE: a) Marketing mismatches may not appeal to families based on their cultural values that is, countries may articulate a meritocratic rationale and it may not resonate [see this example: Patnaik, N. (2023). Teaching development: Critically evaluating ECE for economic growth in India. International Journal of Social Science & Economic Research, 8(08), 2390–2396.]; b) Mismatches between the promises made for ECE and people's experiences may support a backlash or divestment in countries where there is a large uptake (i.e., "my ECE program is low-quality, didn't help my child", etc.); c) The meritocratic promise doesn't only affect individual expectations, it creates a competitive market for ECE in places where it's privatized as families seek to "get ahead" via best ECE like best colleges; and d) ECE teachers who do focus on citizenship and/or social skills may experience pressure from families to "teach hard skills" for individual success (that is, parents as consumers may both believe the marketing & reinforce its expectation on their ECE providers). Fleshing out these real-world implications of your research may help strengthen the rationale for the work you've done & demonstrate that a nation's articulated rationale for investing in ECE may actually, in the long-run, determine its focus and content over time.

We thank Reviewer 2 discussing these relevant implications. We have added a new part in the discussion to reflect on the broader real-world consequences of meritocratic ECE framing. This includes your point about potential mismatches between policy narratives and families’ cultural values (e.g., Patnaik, 2023), the risk of backlash if expectations are unmet, the emergence of competitive ECE markets in privatized settings, and the pressures placed on educators to conform to a more academic or skills-based approach. We believe these additions deepen our analysis of how policy discourse can shape family perceptions and educator practices, with real-world implications to strengthen the rationale of our paper in terms of the implication.

We have highlighted the added paragraph in the revised version of the manuscript.

Best of luck as you continue to revise this paper. It's meaningful and important work, and I'd suggest that you follow-up with a paper on the proliferation of this model & its origins globally.

We are very grateful for your encouraging words and your suggestion to pursue this line of inquiry further. We hope that Reviewer 2 is satisfied with the changes made to the manuscript.

---

## [Editor Report · Decision Letter 2]

24 May 2025

The meritocracy trap: Early childhood education policies promote individual achievement far more than social cohesion

PONE-D-24-28449R2

Dear Dr. Bobrowicz,

We’re pleased to inform you that your manuscript has been judged scientifically suitable for publication and will be formally accepted for publication once it meets all outstanding technical requirements.

Kind regards,

Daphne Nicolitsas

Academic Editor

PLOS ONE
---

## [Editor Report · Acceptance letter]

PONE-D-24-28449R2

PLOS ONE

Dear Dr. Bobrowicz,

I'm pleased to inform you that your manuscript has been deemed suitable for publication in PLOS ONE. Congratulations! Your manuscript is now being handed over to our production team.

Kind regards,

on behalf of

Dr. Daphne Nicolitsas

Academic Editor

PLOS ONE